# Effect of Molecular Weight and Degree of Substitution on the Physical-Chemical Properties of Methylcellulose-Starch Nanocrystal Nanocomposite Films

**DOI:** 10.3390/polym13193291

**Published:** 2021-09-27

**Authors:** Qian Xiao, Min Huang, Xiaolan Zhou, Miaoqi Dai, Zhengtao Zhao, Hui Zhou

**Affiliations:** 1School of Food Science and Technology, Hunan Agricultural University, 410128 Hunan, China; qianxiao@hunau.edu.cn (Q.X.); minhuang11@126.com (M.H.); XiaolanZ90@126.com (X.Z.); MiaoqiD98@163.com (M.D.); 2Department of Food Science, University of Guelph, Guelph, ON N1G 2W1, Canada; zhengtaozhao85@gmail.com

**Keywords:** methylcellulose films, starch nanocrystals, molecular weight, degree of substitution, physicochemical properties

## Abstract

This research studied the effect of molecular weight (M_w_) and degree of substitution (DS) on the microstructure and physicochemical characteristics of methylcellulose (MC) films with or without SNC. The M_w_ and DS of three types of commercial MC (trade name of M20, A4C, and A4M, respectively) were in the range of 0.826 to 3.404 × 10^5^ Da and 1.70 to 1.83, respectively. M_w_ significantly affected the viscosity of methylcellulose solutions as well as the microstructure and tensile strength of methylcellulose films, while DS had a pronounced effect on their oxygen permeability properties. The incorporation of 15% (*w/w*) SNC resulted in the efficient improvement of tensile strength, water, and oxygen barrier properties of films, particularly for the A4C nanocomposite films. The results from SEM and FTIR illustrated that relatively homogenous dispersion of SNC was distinguished in A4C-15% (*w/w*) SNC films. Furthermore, microstructures of MC-SNC nanocomposite films were strongly dependent on both M_w_ and DS of MC. This work offers a convenient and green method to fabricate MC-based nanocomposite films with desirable mechanical, light, oxygen, and water vapor barrier properties.

## 1. Introduction

Synthetic petroleum-based plastics have dominated the food packaging industry, since they are versatile, flexible, durable, and low cost. Despite these advantages, extensive use of plastic packaging materials has led to serious environment problems, due to their non-biodegradability. Biopolymer films, produced from polysaccharide, protein, lipid, and their blends, have received considerable attention as promising candidates for food packaging [1]. According to the Market Data Forecast, the global edible packaging market was valued to be USD 727.6 million in 2021 and is expected to expand at a compound average growth rate of 6.2%, reaching USD 1.1 billion by 2026 [2].

Cellulose, the most abundant natural polymer from renewable sources, is a linear biopolymer of D-glucose rings linked by β-1,4 glycosidic bonds [3]. Methylcellulose (MC) is a water-soluble cellulose derivative, and its hydroxyl groups of the anhydroglucose repeat units are partially substituted by hydrophobic methoxy groups. This substitution pattern endows MC with excellent thermal gelation and film-forming characteristics compared with cellulose [4]. Up to now, it has been demonstrated that molecular weight (M_w_) and degree of substitution (DS) of MC play critical roles in its physicochemical properties, i.e., water solubility, thermal stability, thickening, and gel- and film-forming properties [5]. Commercial MC is usually produced with an average DS value ranging from 1.4 to 1.9 mol of OCH_3_ per mol of AGU unit, which can readily dissolve in water at low temperatures to form homogeneous film-forming solutions [6]. MC films display efficient oxygen and lipid barrier properties; however, their weak mechanical strength and water vapor barrier capability have limited their application [7].

Currently, the incorporation of polysaccharide nanofillers (such as cellulose, chitin, and starch nanocrystals) into biopolymers to fabricate bio-nanocomposite films has gained considerable attention, because of their biodegradability and outstanding physical and functional properties [8]. Starch nanocrystals (SNC) are crystalline platelets produced from the disruption of the semi-crystalline structure of starch granules by the acid hydrolysis of amorphous parts [9]. Researchers have studied the reinforcement effects of SNC in biopolymer films produced from incorporating SNC into carboxymethyl chitosan, pullulan, starch, pea starch, thermoplastic starch, amaranth protein, and soy protein isolate [10,11,12,13,14,15]. All these studies demonstrated that these biopolymer-SNC nanocomposite films exhibited improved mechanical, oxygen and water vapor barrier, surface hydrophobicity, and/or thermal properties in comparison with the pure biopolymer films. Although the improvement effects of SNC on the biopolymer films have been investigated, information on the physicochemical properties of methylcellulose-SNC films has not been reported yet.

In our previous study, SNC from waxy rice starch displayed a square-like platelet morphology with a diameter distribution range from 85.24 to 182.48 nm, while it showed an A-type crystalline structure with a highly crystalline nature [16]. Thus, this SNC could potentially be the promising nanofillers for reinforcement of polysaccharide films. Herein, the first objective was to analyze the M_w_ and DS of three types of commercial MC by using high-performance size exclusion chromatography-angle laser light scattering-refractive index detector (HPSEC-MALL-RI) and gas chromatography (GC). The second objective of the research was to investigate the influence of M_w_ and DS of MC on the morphology and physicochemical characteristics of the MC films and nanocomposite films reinforced with SNC from waxy rice starch.

## 2. Materials and Methods

### 2.1. Materials

Three MC samples, with trade name of M20 (Sinopharm Chemical Co., Ltd., Shanghai, China), A4C, and A4M (Ashland Co., Shanghai, China), were used as received from the manufacturers. Waxy rice starch SNC was prepared using the method reported by our previous study [16]. Sulfuric acid, NaNO_3_, methylbenzene, o-xylene, adipic acid, hydroiodic acid, and methyl iodide (Aladdin Bio-Chem Technology Co., Ltd., Shanghai, China) were used as received from the manufacturers.

### 2.2. Structural and Characterization of Methylcellulose

M_w_ distributions of MC were determined using HPSEC-MALLS-RI (Wyatt Technology Corp., Santa Barbara, CA, USA) as described by Li, Shen, Lyons, Sammler, Brackhagen, and Meunier (2016) [17]. MC powder (100 mg) was dissolved in 1 mL 0.1 M NaNO_3_ solution, filtered through a 0.45 µm filter, and then injected into an HPSEC-MALLS-RI system. The eluting procedure was performed at 0.4 mL/min within ShodexTM OHpak SB-803 HQ, OHpak SB-804 HQ, and SB-805 HQ columns (Showa Denko K.K., Tokyo, Japan) in series.

The methoxy content of MC was estimated according to the United States Pharmacopeia Convention with slight modification [18]. The methylbenzene in o-xylene (30 mg/mL) was used as internal standard. To prepare sample solution, 0.065 g MC was placed in a reaction vial with 0.065 g of adipic acid, 2 mL of internal standard solution, and 57% hydroiodic acid, followed by thoroughly mixing at 130 °C for 20 min. In another vial, 0.135 g of adipic acid, 4 mL of the internal standard solution, and hydroiodic acid were added and sealed. After that, 90 μL of methyl iodide was added and the reaction vial was shacked well. Finally, each 2 μL of sample solution and standard solution were determined using GC equipped with a flame ionization detector (FID) (GC-2010 Pro, Shimadzu, Tokyo, Japan). Based on the methoxy content of MC, the DS value was calculated by Equation (1) [19]:(1)$$DS=\frac{162\times OCH_{3}}{31-\left(14\times OCH_{3}\right)}$$
where *OCH*_3_ is the methoxy content of MC, and 162, 31, and 14 account for molecular weight of AGU units of cellulose, methoxy content, and the net increase in molecular weight of the AGU unit for each methoxy group substituted, respectively.

### 2.3. Viscosity of Film-Forming Solutions

Three MC solutions were prepared by mixing 1.5 g of MC powder into 100 mL of redistilled water, then stirring for 3 h at 60 °C to form homogeneous and transparent solutions. The MC-SNC film-forming solutions were prepared as follows: 1.5 g of MC powder was mixed into 85 mL of redistilled water and then stirred for 5 h at 60 °C to form homogeneous solutions. Aliquots of 15 mL of SNC suspension (1.5 g/100 mL) were homogenized in ice bath using an Ultra Turrax T25 homogenizer at 30,000 rpm (10 min) and then added slowly to prepared MC solutions. The mixtures were mixed thoroughly by using a magnetic stirrer (150 rpm) at 25 °C for 1 h. Based on the weight percentage of SNC/MC ratio, the resulting solutions were referred to as M20-15% SNC, A4C-15% SNC, and A4M-15% SNC, respectively. Viscosity of MC solutions was detected by a rotational rheometer (Kinexus Rheometer, Malvern Instruments. Inc., Shanghai, China) equipped with a 60 mm diameter parallel-plate geometry. Flow behaviors of samples were fitted by power-law equation (Equation (2)):(2)$$\sigma =K\gamma^{n}$$
where σ, γ,
*K*, and *n* are shear stress, Pa, shear rate, s^−1^, consistency index, Pa·sn, and flow behavior index, respectively.

### 2.4. Preparation of Films

Preliminary experiments in MC-based films containing SNC concentrations in the range 5–25% (*w/w*) suggested that addition of 15% (*w/w*) SNC into MC resulted in formation of the coherent and relatively strong films. Here, the MC and MC-SNC solutions as outlined in Section 2.4 were deaerated using a vacuum pump (YC7134, Suli Instruments Co. Ltd., Wenlin, China) and then casted on glass molds. After that, the film-forming solutions were dried in an environmental chamber (GZ-150-HSII, Kezhi Instruments Co. Ltd., Shaoguan, China) at 48 °C and 55% RH for 8 h.

### 2.5. Scanning Electron Microscope (SEM)

The cross-section morphology of film was observed by a scanning electron microscope (SU8020, Hitachi Ltd., Tokyo, Japan). Films were coated with gold under vacuum and then scanned with an accelerating beam voltage of 3 kV.

### 2.6. Fourier Transform Infrared Spectroscopy (FTIR)

FTIR spectra were determined by using an ATR-FTIR spectrometer (FTIR-8400S, Shimadzu Corporation, Tokyo, Japan). All spectra were collected from 4000 to 700 cm^−1^ with a resolution of 4 cm^−1^ at 25 °C.

### 2.7. X-ray Diffraction (XRD)

The XRD diffraction of films was characterized by Bruker D8 Advance model X-ray diffractometer (Brucker GmbH, Karlsruhe, Germany) equipped with Cu Kα radiation. The results were collected in the range from 2θ = 4° to 2θ = 40° at 2° min^−1^ scan rate.

### 2.8. Thermogravimetric Analyzer (TGA)

TGA of films was determined on a TGA instrument (STA 449 F5/F3 Jupiter, Netzsch, Hanau, Germany). The samples were heated under a nitrogen atmosphere at 10 mL/min flow rate at a heating rate of 10 °C/min.

### 2.9. Light Transmission

Light transmission of films was measured by using a UV-2600 spectrophotometer (Shimadzu Corporation, Tokyo, Japan). The films were cut into a rectangular specimen and placed in the spectrophotometer cell. An empty compartment was used as a reference in the measurements. The light barrier properties of the film samples were measured by scanning the samples at wavelengths between 300 and 800 nm using the spectrophotometer.

### 2.10. Mechanical and Barrier Properties

Tensile strength (TS) and elongation at break (EAB) of film specimens were analyzed by an Electronic Universal Testing Machine (Jiangsu Kaiyuan Testing Equipment Co., Ltd., Nanjing, China) based on ASTM standard method D882-02 [20].

Water vapor permeability (WVP) of film specimens was determined using ASTM E96 method [21]. Additionally, oxygen transmission rate (OTR, mL·m^−2^·day^−1^) of films was obtained by oxygen transmission rate tester (BTY-B1, Labthink Inc., Jinan, China) at 25 °C and 55% RH, according to ASTM D3985-02 [22].

### 2.11. Statistical Analysis

Analysis of variance (ANOVA) was analyzed by using SPSS software (SPSS Inc., Chicago, IL, USA).

## 3. Results and Discussions

### 3.1. Characteristics of Methylcellulose

The weight average molar mass (M_w_), polydispersity index (M_w_/M_n_), radius of gyration (R_z_), methoxy content, and DS of three types of MC are summarized in Table 1. The M_w_ values of M20, A4C, and A4M were 0.826 × 10^5^, 1.329 × 10^5^, and 3.404 × 10^5^ g/mol, respectively. The M_w_/M_n_ value of M20 was obviously higher than that of A4C and A4M, implying that M20 showed a broad M_w_ distribution among the tested MC samples [23]. The R_z_ of MC increased with an increase in its M_w_ value: A4M displayed the largest gyration radius (102 nm), and M20 had the smallest (53.0 nm). As shown in Table 1, the methoxy content of M20, A4C, and A4M was 29.1, 28.4, and 30.2%, respectively. Correspondingly, the resulting DS values for M20, A4C, and A4M were 1.75, 1.70, and 1.83, respectively.

### 3.2. Viscosity of Film-Forming Solutions

The viscosity of MC and MC-SNC solutions is presented in Figure 1. As shown in Figure 1a, no significant change in the viscosity value of 1.5% (*w/w*) M20 solutions was observed with increasing shear rate from 0.1 to 100 1/s, indicating that the 1.5% (*w/w*) M20 solution showed Newtonian flow behavior. In contrast, A4C and A4M solutions showed the shear thinning behavior at the same concentration. Moreover, the viscosity increased dramatically with their M_w_ increasing, probably due to the formation of larger aggregates in MC aqueous solution with increasing its M_w_ [24]. For MC-15% (*w/w*) SNC solutions, *n* values of all the samples were significantly lower than unity, and the viscosity was obviously higher than that of pure MC solutions within the tested shear rate range. This result indicated that SNC acted as an effective thickening agent in MC solutions. The increased viscosity could be attributed to the flow-impeding effect induced by the presence of SNC in the biopolymer matrix [25]. Since the particle size of SNC is larger than that of the chain segments of MC biopolymers, SNC restrained the shear flow of MC macromolecules, which resulted in an increase in viscosity [26]. Similar behaviors were reported for alginate-cellulose nanocrystal and starch-cellulose nanocrystal solutions [26,27].

### 3.3. SEM

SEM photographs of cross-sections of MC films and MC-SNC nanocomposite films are presented in Figure 2. M20 films showed a porous network structure (Figure 2a), while A4C and A4M films exhibited a fibrillar network structure along with nonuniform holes (Figure 2b,c). Additionally, the hole size of M20 films in the range from 11 to 120 nm was significantly larger than that of A4C (from 8 to 78 nm) and A4M films (from 5 to 35 nm) (Appendix A). This phenomenon could mainly be related to the difference in their M_w_. MC chains have a tendency to form fibrils in aqueous solutions above 40 °C, and the increase in M_w_ leads to the production of longer fibrils [28]. Therefore, the MC with higher M_w_ assembled into the more connected fibrillar network with a smaller hole size during the film formation.

By the incorporation of SNC to MC matrix, the cross-section microstructure became markedly different. As shown in Figure 2d, the irregular and micron-sized agglomerations of SNC were obviously stacked in the M20-15% (*w/w*) SNC nanocomposite films, probably due to their lowest M_w_ value. As mentioned in Section 3.2, Newtonian flow behavior was observed for the 1.5% (*w/w*) M20 solution alone. This indicated that chains of M20 were contacting each other to form a transient network that was disrupted readily, thereby forming large agglomerations of SNC during film formation (as schematically shown in Figure 3a). In contrast, the relatively homogenous dispersion of SNC was distinguished in the A4C-15% (*w/w*) SNC film, except individually or in small aggregates at the A4C matrix (Figure 2e). This indicated that SNCs have better compatibility with A4C than M20 and A4M, which was probably ascribed to the favorable interfacial interactions between the SNC and A4C matrix. In A4M-15% (*w/w*) SNC films, the SNCs were partially aggregated to form the submicrometer-sized aggregations, which were not uniformly distributed throughout the nanocomposite films (Figure 2f). This could be due to their high DS value, which led to the lower capacity to fabricate the intermolecular interactions between SNC and A4M, promoting the aggregation of SNC. Based on the abovementioned SEM results, the schematic of the distribution of SNC in the three-dimensional network fabricated by MC chains could be provided, as shown in Figure 3.

### 3.4. ATR-FTIR

FTIR spectra for MC films and MC-SNC nanocomposite films are summarized in Figure 4. The band around 3441 cm^−1^ in all samples was attributed to O-H stretching vibration [16]. The bands at around 2973, 2932, 2903, and 2837 cm^−1^ were due to the symmetric and asymmetric stretching vibration of C-H bonds, while the bands around 1420 and 1374 cm^−1^ were determined by the deformation vibrations of CH_2_ and bending vibration of CH groups, respectively, due to the presence of the CH_3_ groups of MC and CH and CH_2_ groups of SNC [29]. As presented in Figure 4A, the 900–1150 cm^−1^ region of the spectra displayed several overlapping bands. To accurately study the vibration bands and their frequency shift, a second derivative was shown in Figure 4B,C. For pure MC films, six bands at around 1112, 1074, 1054, 1025, 1009, and 945 cm^−1^ were revealed (Figure 4B). The absorbance bands at around 1112, 1074, 1054, 1025, and 945 cm^−1^ were related to the asymmetric stretching of C-O-C of the glucoside bridge, symmetric stretching of C-O for primary alcohol, stretching vibration of C-O, vibration of the OCH_3_ groups, and asymmetric stretching of the glucose ring for MC, respectively [30]. For the SNC spectrum, the absorbance bands at 1105, 1075, and 1059 cm^−1^ were attributed to the stretching vibration of C-C, bending vibration of C-O-H, and stretching vibration of C-OH, respectively. The bands at around 1048, 1013, and 995 cm^−1^ were due to the C-O-H, deformation vibration of C-OH, and bending vibration of C-O-C in the glucose ring, respectively, while 955 cm^−1^ was due to the coupled vibration of C-C and C-O [16,31]. Thus, a new band at around 1048 cm^−1^ was observed for all MC-SNC nanocomposite films (Figure 4C), which was associated with the ordered double helix structure of SNC [16]. As shown in Figure 4B,C, the absorbance band at around 1025 cm^−1^ significantly shifted to a lower wavenumber (around 1020 cm^−1^) with the addition of SNC into MC films. This observation implied that the OCH_3_ groups for MC have a tendency to form a hydrogen bond with SNC in the nanocomposite films, which may be responsible for the improved mechanical strength of MC-SNC films.

### 3.5. XRD

X-ray diffractograms of MC films and MC-SNC films are displayed in Figure 5. Two broad diffraction peaks at 2θ around 7.8° and 20.5° were presented in the diffractogram of M20 films, indicating that M20 showed the amorphous state. A similar result has been reported for the same type of commercial MC [32]. For M20-SNC nanocomposite films, besides the aforementioned peaks shown in the diffractogram of M20 films, a new peak at around 15.05° appeared, which was due to the crystalline structure of SNC [16]. In contrast, both diffractograms of A4C and A4M films displayed a sharp peak at 2θ = 8.02° and a broad peak at 2θ = 20.5°, indicating that A4C and A4M films showed a typical semi-crystalline state. Similar XRD patterns have been reported by Kumar et al. (2012). Moreover, the position of the sharp peak for A4C (around 8.02°) was slightly higher than that for A4M (around 7.81°), indicating the difference in DS of A4C and A4M. Moreover, the projection of the methyl groups in MC is related to an increase in the interfibrillar distance [33]. Incorporating 15% (*w/w*) SNC into A4C and A4M films resulted in no significant difference in diffraction patterns of nanocomposite films. This could be attributed to the high crystallinity degree of A4C and A4M compared to that of SNC, resulting in the predominant role of the biopolymer in the nanocomposite films.

### 3.6. TGA

TGA and derivative thermogravimetry (DTG) curves of MC films and MC-SNC nanocomposite films are shown in Figure 6. As presented in Figure 6a, two weight loss stages were observed for pure MC films, as reported by Tunç and Duman (2010) [34]. The first stage (below 100 °C) related to the evaporation of absorbed water, while the second stage was attributed to the degradation of MC chains [7]. Moreover, thermal depolymerization of A4C and A4M films started later than M20 films, which was due to the low M_w_ of M20. Similar results have been reported for chitosan with a different M_w_ [35]. However, the thermal decomposition of MC-SNC films occurred in three main stages. The second and third stages were related to the degradation of sulphate groups of SNC and the decomposition of macromolecules, respectively [16]. Furthermore, M20-15% (*w/w*) SNC films showed a lower degradation temperature at the second and third stages of weight loss (218.8 and 357.6 °C, respectively) compared to other nanocomposite films (Figure 6b). According to Kassab et al. (2019), the increased degradation temperature of cellulose nanocrystals/κ-carrageenan nanocomposite films at the second stage was due to the strong interfacial interaction [36]. Therefore, in this study, the reason for the lower degradation temperature at the second stage could be ascribed to the formation of larger agglomerations of SNC in M20-15% (*w/w*) SNC films, which decreased the interfacial interaction between SNC and M20. In comparison with other nanocomposite films, the lower degradation temperature of M20-15% (*w/w*) SNC films at the third stage could be attributed to the lower M_w_ and amorphous state of M20. The temperature at 10% weight loss (T_10%_) and char yield of MC films and MC-SNC nanocomposite films are summarized in Table 2. Among the tested MC films, M20 showed the lowest T_10%_ values (292.2 °C), probably due to lowest molecular weight of M20. A similar trend has been observed for the char yield of MC films. For MC-SNC nanocomposite films, all samples exhibited lower T_10%_ values than those of MC films, indicating that the addition of SNC to MC films resulted in the decreased thermal stability of MC films. However, the char yield of M20-15% (*w/w*) SNC, A4C-15% (*w/w*) SNC, and A4M-15% (*w/w*) SNC was 16.25, 16.71, and 18.49%, respectively, which was higher than that of pure MC films.

### 3.7. Light Transmission

Figure 7 shows the transmittance of MC films and MC-SNC nanocomposite films in the wavelength range of 250–800 nm. In the visible region, the transmittance of M20 films was slightly higher than that of A4C and A4M films, while no difference was observed for the transmittance of M20, A4C, and A4M films in the UV region. Overall, the addition of SNCs improved the barrier to UV and visible light of the nanocomposite films. The transparency of MC-15% (*w/w*) SNC films strongly depended on the dispersion of SNCs within the biopolymer matrix. Any aggregation of SNCs in the MC matrix would increase the scattering effect, resulting in reduced film transparency [37]. Similar results have been reported for poly(butylmethacrylate)-SNC, cassia gum-carboxylated cellulose nanocrystal, and carboxymethyl cellulose-cellulose nanocrystal films [26,37]. It was noticed that the transmittance of A4C-15% (*w/w*) SNC films was significantly higher than that of M20 and A4M blend films with 15% (*w/w*) SNC in the tested wavelength region, probably due to the relatively good distribution of SNCs in the A4C matrix. This is consistent with the SEM findings as mentioned in Section 3.3. Although the larger agglomerations of SNC were observed in M20-15% (*w/w*) SNC films, no significant difference was observed for the transmittance in the UV and visible region (*p* < 0.05) between M20-15% (*w/w*) SNC and A4M-15% (*w/w*) SNC films. This could be attributed to the difference of the M_w_, M_w_ distributions, DS, and crystalline structure of M20 and A4C. Furthermore, the inclusion of SNC significantly decreased the rate of transmittance compared to pure MC films in the UV region, implying that this nanocomposite film could be potentially utilized as packaging material with a UV light barrier property.

### 3.8. Mechanical Properties

The mechanical properties of MC films and MC-SNC nanocomposite films are summarized in Table 3. A4M films exhibited the highest TS (68.31 MPa), YM values (890.98 MPa), and stretchability (8.4%) among the tested MC films. This could be attributed to the highest molecular weight of A4M, implying the higher level of physical entanglement arising from the longer chains of A4M compared to M20 and A4C. A similar result has been reported for pullulan and HPMC films with different M_w_ [38,39]. For MC-SNC nanocomposite films, A4C-15% (*w/w*) SNC and A4M-15% (*w/w*) SNC films showed significantly higher TS values than pure MC films. However, no obvious difference in TS values was observed between M20 and M20-15% (*w/w*) SNC films. This could be attributed to the formation of a micron-sized agglomeration of SNC in the M20 matrix, which was consistent with the previous electron scanning microscopy findings (Figure 2d). On the contrary, YM values of nanocomposite films were significantly higher than those of pure MC films, due to the high rigidity exerted by the SNC. As shown in Table 3, all EAB values decreased remarkably after incorporating 15% (*w/w*) SNC into MC films, probably due to the rigid behavior of SNC and their interfacial interactions with the biopolymer, which prevent the motion of the macromolecular chains of the biopolymer [40]. Note that A4C-15% (*w/w*) SNC films exhibited the highest TS value (75.49 MPa) among the tested samples due to the homogenous distribution of SNC accompanied by the favorable interfacial interactions between the SNC and A4C matrix.

### 3.9. Barrier Properties

No significant difference was observed for WVP values of M20, A4C, and A4M films due to their hydrophilic properties, while the addition of 15% (*w/w*) SNC into MC films resulted in a decrease in the WVP value (Table 3). This could be due to generating a tortuous path for water molecules by the incorporation of SNC into the biopolymer matrix [14]. Among the tested films, A4C-15% (*w/w*) SNC films displayed the lowest WVP value (4.79 × 10^−4^ g·m·Pa^−1^·h^−1^·m^−2^), probably due to the good distribution of SNC in the A4C matrix, which led to the formation the longer diffusion path for water molecules [14]. Similar phenomena have been reported for amaranth protein-SNC films and carboxymethyl cellulose/starch-cellulose nanocrystal films [12].

M20 films showed the highest OTR value (275.9 cm^3^·m^−2^·day^−1^) among the tested MC films (Table 3). This could be a result of the more porous structure of M20 films compared to A4C and A4M films, which might facilitate the permeation of oxygen. However, for A4C and A4M films, although the hole size of A4M films was smaller than that of A4C films, the OTR value (248.1 cm^3^·m^−2^·day^−1^) was significantly higher than that of A4C (206.3 cm^3^·m^−2^·day^−1^). This could be related to the higher DS value of A4M, which led to the higher capacity of interaction with oxygen molecules compared to A4C films. Regarding the OTR results of MC-SNC nanocomposite films, it was expected that the addition of SNC could led to the decrease of the OTR values, due to the capacity of these platelet-like nanocrystals to produce a tortuous path for oxygen molecules, decreasing the diffusivity [41]. Similar results have been reported for thermoplastic starch-SNC, whey protein isolate, and starch films reinforced with CNC [42,43]. These observations indicated that MC-SNC nanocomposite films could potentially be a packaging material for meat, poultry, and/or seafood products due to their desirable oxygen barrier properties.

## 4. Conclusions

In this study, the morphology and physico-chemical properties of MC films are significantly influenced by M_w_, DS, and the incorporation of SNC. M_w_ significantly affected the microstructure and tensile strength of methylcellulose films. For example, M20 films showed a porous network structure, while A4C and A4M films exhibited a fibrillar network structure along with nonuniform holes. A4M films exhibited the highest TS, YM values, and stretchability among the tested MC films. On the other hand, the OTR value of A4M was 248.1 cm^3^·m^−2^·day^−1^, which was remarkably higher than that of A4C (206.3 cm^3^·m^−2^·day^−1^), probably due to the fact that DS had a pronounced effect on their oxygen permeability properties. Nanocomposite films incorporated with 15% (*w/w*) SNC had a higher TS value and lower transmittance, WVP, and OTR values than pure MC films, especially for the A4C-SNC nanocomposite films. Furthermore, both M_w_ and DS values of MC significantly affected the microstructure of nanocomposite films, while M_w_ had a pronounced effect on their EAB values. The fabrication of MC-SNC nanocomposite films provides a convenient, green, and environmentally friendly route to develop packaging materials with desirable mechanical, light, oxygen, and water vapor barrier properties, which could potentially be applied to meat, poultry, and/or seafood products.

## Figures and Tables

**Figure 1 polymers-13-03291-f001:**
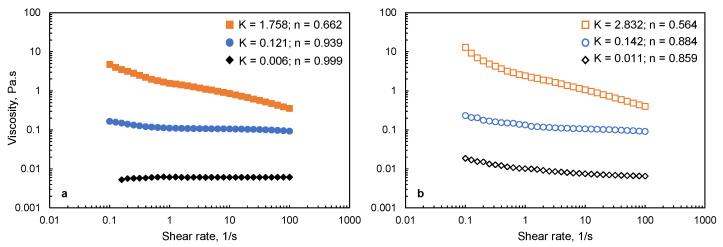
Apparent viscosity of (**a**) MC film-forming solutions (♦) M20; (●) A4C; (■) A4M; and (**b**) MC-SNC film-forming solutions (◊) M20-15% SNC; (○) A4C-15% SNC; (□) A4M-15% SNC.

**Figure 2 polymers-13-03291-f002:**
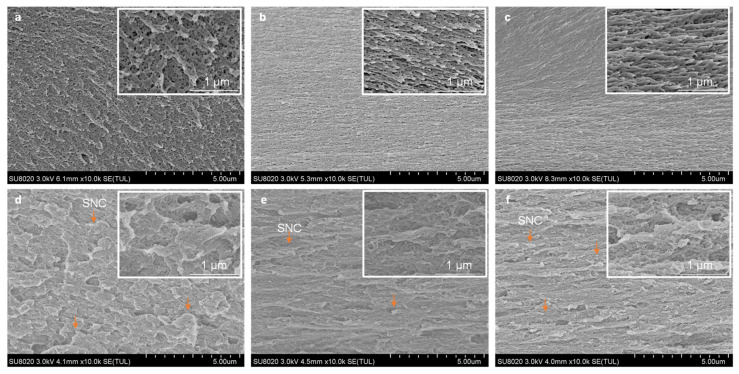
SEM image of cross-section for MC films and MC-SNC nanocomposite films: (**a**) M20; (**b**) A4C; (**c**) A4M; (**d**) M20-15% SNC; (**e**) A4C-15% SNC; (**f**) A4M-15% SNC.

**Figure 3 polymers-13-03291-f003:**
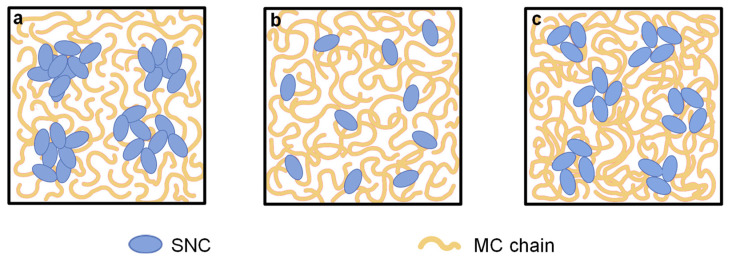
A schematic illustrating the distribution of SNC in MC matrix: (**a**) M20-15% SNC; (**b**) A4C-15% SNC; (**c**) A4M-15% SNC.

**Figure 4 polymers-13-03291-f004:**
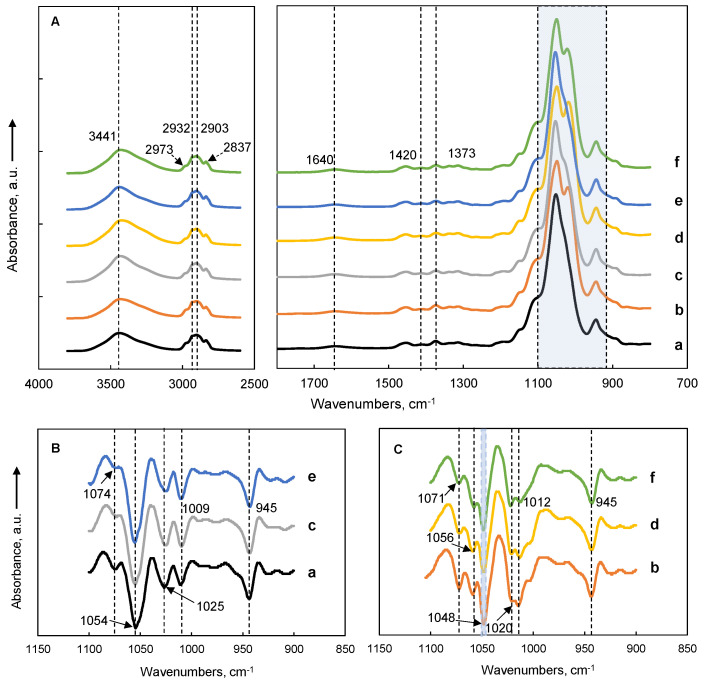
Origin ATR-FTIR spectra (**A**) and second derivative spectra in the range of 900–1100 cm^−1^ (**B**,**C**) of MC films and MC-SNC nanocomposite films: (a) M20; (b) M20-15% SNC; (c) A4C; (d) A4C-15% SNC; (e) A4M; (f) A4M-15% SNC.

**Figure 5 polymers-13-03291-f005:**
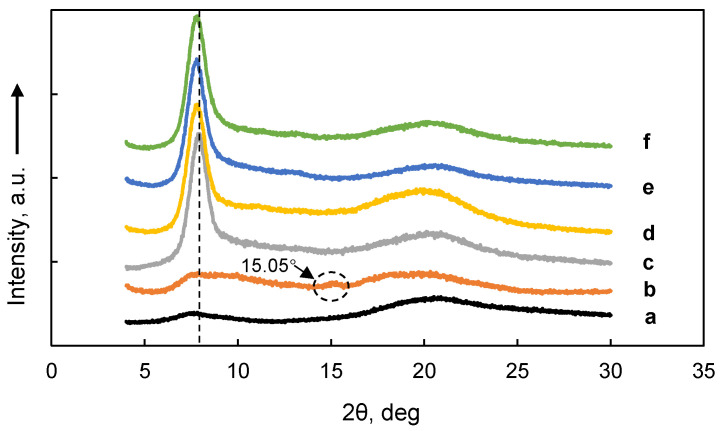
X-ray diffractograms of MC films and MC-SNC nanocomposite films: (a) M20; (b) M20-15% SNC; (c) A4C; (d) A4C-15% SNC; (e) A4M; (f) A4M-15% SNC.

**Figure 6 polymers-13-03291-f006:**
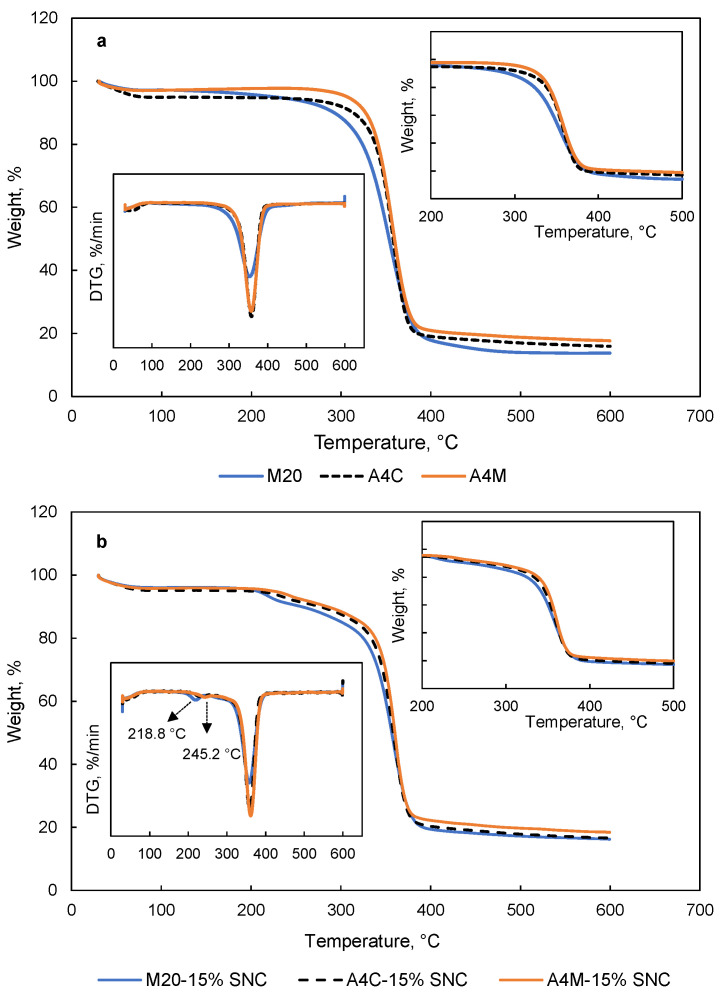
TGA and DTG curves of (**a**) MC films and (**b**) MC-SNC nanocomposite films.

**Figure 7 polymers-13-03291-f007:**
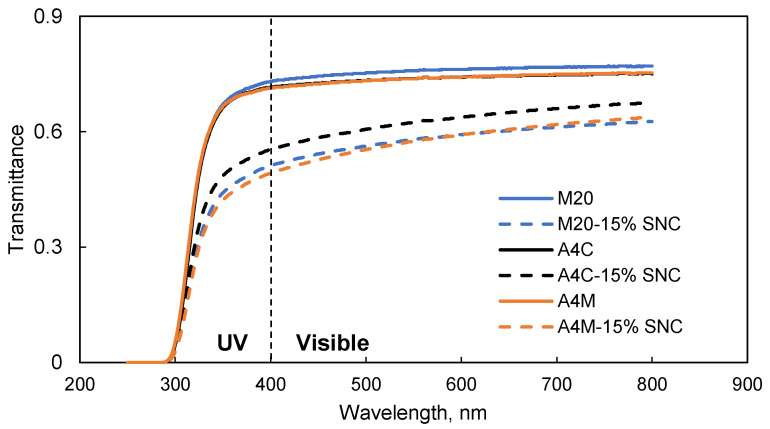
Light transmittance of MC films and MC-SNC nanocomposite films.

**Table 1 polymers-13-03291-t001:** The weight average molar mass (M_w_), polydispersity index (M_w_/M_n_), radius of gyration (R_z_), methoxy content, and DS of MC.

Samples	M_w_ × 10^5^, Da	M_w_/M_n_	R_z_, nm	Methoxy Content, %	DS
M20	0.826 ± 0.02 ^c^	3.45 ± 0.07 ^a^	53.0 ± 1.71 ^c^	29.1 ± 0.26 ^b^	1.75 ± 0.04 ^b^
A4C	1.329 ± 0.01 ^b^	2.92 ± 0.06 ^b^	57.7 ± 2.07 ^b^	28.4 ± 0.45 ^c^	1.70 ± 0.06 ^c^
A4M	3.404 ± 0.06 ^a^	2.76 ± 0.08 ^b^	102.0 ± 1.91 ^a^	30.2 ± 0.32 ^a^	1.83 ± 0.03 ^a^

a–c Least square means with different letters within M20, A4C and A4M are significantly different (*p* < 0.05).

**Table 2 polymers-13-03291-t002:** Thermal properties of MC films and MC nanocomposite films.

Samples	T_10%_, °C	Char Yield, %	T_d2_, °C	T_d3_, °C
M20	292.2 ± 4.27 ^c^	13.79 ± 2.32 ^b^	NA	354.4 ± 4.63 ^b^
A4C	313.3 ± 5.83 ^b^	15.91 ± 2.16 ^a^	NA	358.0 ± 2.63 ^a^
A4M	327.4 ± 3.36 ^a^	17.66 ± 1.98 ^a^	NA	357.4 ± 3.79 ^a^
M20-15% SNC	256.0 ± 3.19 ^C^	16.25 ± 2.07 ^A^	218.8 ± 2.99 ^B^	357.6 ± 2.75 ^A^
A4C-15% SNC	276.4 ± 5.29 ^B^	16.71 ± 1.85 ^A^	241.4 ± 3.58 ^A^	358.8 ± 3.18 ^A^
A4M-15% SNC	285.0 ± 2.91 ^A^	18.49 ± 2.19 ^B^	245.2 ± 3.21 ^A^	360.5 ± 3.85 ^B^

T_10%_: temperature at 10% weight loss; T_d2_: degradation temperature at the second stage of weight loss; T_d3_: degradation temperature at the third stage of weight loss. a–c Least square means with different letters within MC films are significantly different (*p* < 0.05). A–C Least square means with different letters within MC-SNC nanocomposite films are significantly different (*p* < 0.05).

**Table 3 polymers-13-03291-t003:** Tensile strength, Young’s modulus, elongation at break, water vapor permeability, and oxygen transmission rate of MC film with and without SNC.

Samples	TS, MPa	YM, MPa	EAB, %	WVP × 10^−4^ g·m·Pa^−1^·h^−1^·m^−2^	OTR, cm^3^·m^−2^·day^−1^
M20	38.11 ± 5.62 ^c^	651.71 ± 25.62 ^c^	4.71 ± 0.50 ^c^	6.22 ± 0.22 ^a^	275.92 ± 4.31 ^a^
A4C	61.15 ± 3.36 ^b^	757.69 ± 34.66 ^b^	5.77 ± 0.49 ^b^	6.09 ± 0.25 ^a^	206.30 ± 6.06 ^b^
A4M	68.31 ± 4.21 ^a^	890.98 ± 46.89 ^a^	8.42 ± 0.68 ^a^	6.28 ± 0.21 ^a^	248.06 ± 2.22 ^c^
M20-15% SNC	40.43 ± 2.96 ^C^	757.72 ± 19.07 ^C^	1.93 ± 0.56 ^C^	6.07 ± 0.26 ^A^	204.17 ± 3.71 ^A^
A4C-15%SNC	75.49 ± 4.49 ^A^	1050.6 ± 56.67 ^B^	4.68 ± 0.22 ^B^	4.79 ± 0.25 ^C^	156.04 ± 3.82 ^B^
A4M-15%SNC	72.99 ± 3.35 ^B^	1106.5 ± 24.25 ^A^	7.66 ± 0.37 ^A^	5.65 ± 0.14 ^B^	151.84 ± 1.92 ^C^

a–c Least square means with different letters within MC films are significantly different (*p* < 0.05). A–C Least square means with different letters within MC-SNC nanocomposite films are significantly different (*p* < 0.05).

## Data Availability

All pertinent data are included in the body of the manuscript.

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
