# Peer review of "Effect of Molecular Weight and Degree of Substitution on the Physical-Chemical Properties of Methylcellulose-Starch Nanocrystal Nanocomposite Films"

_polymers, 2021, doi:10.3390/polym13193291_

Round 1

Reviewer 1 Report

Manuscript has many characterization methods of polymers science

- In the abstract was difficult to understand M20, A4C and A4M, because the readers will no information about these abbreviations

- 2.1. Materials “All other chemicals” include information name of chemical compounds, solvents, etc.

- Materials, include information about purification materials or include sentence “used as received”

- Lines 159 and 160 change this sentence “A4C and A4M were 1.75, 1.70 and 1.83” for “A4C and A4M were 1.75, 1.70 and 1.83 resectively”

- Lines 226 and 227 “For pure MC films, six bands at around 1112, 1074, 1054, 1025, 1009 and 945 cm-1 were revealed (Fig. 4B)” include chemical groups or vibrations for these numbers

- Figure 4 change “Absorbance” for Absorbance (a.u.)”

- Figure 5 change “Intensity” for “Intensity (a.u.)”

- Line 267 define acronymic “DTG”, derivation?

- 3.6. TGA, include information about 10% weight loss and char yield at 600 °C

- Include important results in the conclusion part

- Manuscript has some interesting results but needs to improve discussion for all Figures because some Figures have no discussion

- Manuscript has only one reference from 2021, include more recently references

Author Response

  1. - In the abstract was difficult to understand M20, A4C and A4M, because the readers will no information about these abbreviations

Thank you for your careful review. The abbreviation of M20, A4C, and A4M was according to their trade name. This sentence was revised to “The Mw and DS of three types of commercial MC (trade name of M20, A4C and A4M, respectively) were in the range of 0.826 to 3.404×105 Da and 1.70 to 1.83, respectively”.

  1. - 2.1. Materials “All other chemicals” include information name of chemical compounds, solvents, etc.

Thank you for your careful review. The name of chemical compounds and solvents was added.

  1. - Materials, include information about purification materials or include sentence “used as received”

Thank you for your careful review. Section 2.1 was revised. 

  1. - Lines 159 and 160 change this sentence “A4C and A4M were 1.75, 1.70 and 1.83” for “A4C and A4M were 1.75, 1.70 and 1.83, respectively”

Thank you for your careful review. This sentence was revised.

  1. - Lines 226 and 227 “For pure MC films, six bands at around 1112, 1074, 1054, 1025, 1009 and 945 cm-1 were revealed (Fig. 4B)” include chemical groups or vibrations for these numbers

The absorbance bands at around 1112, 1074, 1054, 1025, and 945 cm-1 were related to the asymmetric stretching of C-O-C of glucoside bridge, symmetric stretching of C-O for primary alcohol, stretching vibration of C-O, the vibration of the OCH3 groups, and asymmetric stretching of glucose ring for MC, respectively. This content was added.

  1. - Figure 4 change “Absorbance” for Absorbance (a.u.)”

Thank you for your careful review. Fig. 4 was revised.

  1. - Figure 5 change “Intensity” for “Intensity (a.u.)”

Thank you for your careful review. Fig. 5 was revised.

  1. - Line 267 define acronymic “DTG”, derivation?

Thank you for your careful review. DTG is the abbreviation for derivative thermogravimetry. This content was added.

  1. - 3.6. TGA, include information about 10% weight loss and char yield at 600 °C

Thank you for your careful review. The temperature at 10% weight loss and char yield at 600 °C was added in Table 2.

  1. - Include important results in the conclusion part

Thank you for your careful review. The conclusion was revised.

  1. - Manuscript has some interesting results but needs to improve discussion for all Figures because some Figures have no discussion

Thank you for your careful review. The discussions have been added in Section 3.

  1. - Manuscript has only one reference from 2021, include more recently references

Thank you for your careful review. A part of the references in this manuscript was updated to 2021.

Reviewer 2 Report

The article by Xiao et al. deals with methylcellulose-starch Nanocomposites. The article is well written and topic suitable for the journal. The results are well presented. They have performed a comprehensive characterization of the Nanocomposites by different tecniques. I Believe It can be accepted for publication after revision:

-Higher resolución SEM imagen should be provided

- Enlarge the TGA figure to show the Rangel 250- 500 °C. Lista the degradación temperaturas in a Table

- Enlarge the transmittance plot to show better 250- 750 nm

  • Indicate the Young Modulus of the Nanocomposites. Provide further discussion about mechanical properties
  • Highlight the novelty of the work

Author Response

  1. -Higher resolución SEM imagen should be provided

Thank you for your careful review. The higher resolution image was added in Figure 1.

  1. - Enlarge the TGA figure to show the Rangel 250- 500 °C. Lista the degradación temperaturas in a Table

Thank you for your careful review. The enlarged TGA figure in the range from 250 to 500 °C was added, and the degradation temperature at the second and third stages of MC and nanocomposite films was summarized in Table 2.

  1. - Enlarge the transmittance plot to show better 250- 750 nm

Thank you for your careful review. The transmittance figure was enlarged in the range from 250 to 700 nm.

  1. Indicate the Young Modulus of the Nanocomposites. Provide further discussion about mechanical properties

The Young modulus was added in Table 3, and the discussion of Table 3 was added in Section 3.8.

  1. Highlight the novelty of the work

Thank you for your careful review. The highlight was summarized in the conclusion.

Round 2

Reviewer 2 Report

The manuscript has been improved, is now suitable for publication

Author Response

There are some grammar or formatting errors in lines 77, 101, 162, etc.

Thank you for your careful review. These errors were revised.